# The Inability to Disassemble Rad51 Nucleoprotein Filaments Leads to Aberrant Mitosis and Cell Death

**DOI:** 10.3390/biomedicines11051450

**Published:** 2023-05-15

**Authors:** Tadas Andriuskevicius, Anton Dubenko, Svetlana Makovets

**Affiliations:** Institute of Cell Biology, School of Biological Sciences, University of Edinburgh, Alexander Crum Brown Road, Edinburgh EH9 3FF, UK

**Keywords:** budding yeast, Rad51 nucleoprotein filament disassembly, Rad54 and Srs2, stalled replication forks, mitotic bridges

## Abstract

The proper maintenance of genetic material is essential for the survival of living organisms. One of the main safeguards of genome stability is homologous recombination involved in the faithful repair of DNA double-strand breaks, the restoration of collapsed replication forks, and the bypass of replication barriers. Homologous recombination relies on the formation of Rad51 nucleoprotein filaments which are responsible for the homology-based interactions between DNA strands. Here, we demonstrate that without the regulation of these filaments by Srs2 and Rad54, which are known to remove Rad51 from single-stranded and double-stranded DNA, respectively, the filaments strongly inhibit damage-associated DNA synthesis during DNA repair. Furthermore, this regulation is essential for cell survival under normal growth conditions, as in the *srs2Δ rad54Δ* mutants, unregulated Rad51 nucleoprotein filaments cause activation of the DNA damage checkpoint, formation of mitotic bridges, and loss of genetic material. These genome instability features may stem from the problems at stalled replication forks as the lack of Srs2 and Rad54 in the presence of Rad51 nucleoprotein filaments impedes cell recovery from replication stress. This study demonstrates that the timely and efficient disassembly of recombination machinery is essential for genome maintenance and cell survival.

## 1. Introduction

Homologous recombination (HR) is an integral part of DNA metabolism providing both genome stability and genetic diversity in all domains of life. In eukaryotes, a well-conserved recombinase Rad51 plays a key role in this process [1]. Rad51 polymerises on DNA at the sites of damage forming nucleoprotein filaments which govern the homology search and the strand exchange reaction between the damaged DNA molecule and a homologous donor of intact sequences [1,2,3]. This process is essential for the accurate repair of DNA double-strand breaks (DSBs) and *rad51Δ* mutants have an increased sensitivity to DNA damaging factors, such as ionising radiation, bleomycin, and methyl methanesulfonate (MMS) [4,5,6,7]. More importantly, even under normal growth conditions, Rad51 is required for a faithful lesion bypass at challenged replication forks and post-replicative gaps [8,9,10], and Rad51 deficient cells exhibit accumulation of spontaneous DSBs and higher rates of chromosome loss [4,11,12,13]. In addition to enabling the lesion bypass during replication, Rad51 has been shown to limit the access of nucleases to challenged replication forks, thereby protecting nascent DNA strands from degradation [8,14]. Rad51 is also required for fork reversal, a mechanism which has been implicated in both lesion bypass and the stabilisation of stalled replication forks [15,16]. Although fork reversal in yeast seems to be largely inhibited by the DNA damage checkpoint, it appears to be a routine response to genotoxic insults in human cells [15,17]. While Rad51 is not essential in yeast, the lack of the recombinase leads to lethality in mammalian and avian cells, most likely because of a greater extent of spontaneous DNA damage which comes with the larger genome sizes [11,12,13]. Therefore, Rad51 is important not only for the repair of DSBs but also for minimising their formation during replication.

In agreement with their prominent role in genome stability, Rad51 nucleoprotein filaments are tightly regulated. In *S. cerevisiae*, Rad51 polymerisation is initiated by Rad52 which interacts with the ssDNA-binding protein RPA [18,19]. Rad52 facilitates the replacement of RPA by Rad51 on ssDNA by priming Rad51 filaments, which are stabilised by the Rad55-Rad57 heterodimers [18,20,21,22,23,24,25]. These three proteins, also called Rad51 mediators, are necessary for the formation of stable and functional Rad51 nucleoprotein filaments [26,27,28]. A SWI/SNF translocase Rad54 has also been shown to stabilise the binding of Rad51 to ssDNA in vitro [29,30]. Although it is not essential for nucleoprotein filament formation, Rad51 foci appear to form with delayed kinetics and exhibit decreased robustness in the absence of Rad54 in vivo [31,32,33,34]. Once stable Rad51 filaments are formed, they can invade homologous donor sequences. This step is essential for all the downstream HR pathways [1,3,26] and requires Rad54 which promotes the formation of properly intertwined joint DNA molecule intermediates and stabilises them [26,35,36,37,38,39]. Alternatively, *S. cerevisiae* helicase Srs2 can remove Rad51 from ssDNA, allowing the DNA lesions to be repaired or bypassed by different mechanisms [40,41,42,43,44,45].

The significance of the Rad51 filament formation for genome stability is well understood. However, the accumulating evidence suggests that the filament disassembly might be just as important. The Rad51 binding to its DNA substrates has been shown to inhibit the loading of the DNA polymerase processivity factor PCNA in vitro [46,47]. PCNA is loaded onto dsDNA at the dsDNA–ssDNA junction and this reaction is greatly stimulated by RPA [1,46,47,48,49,50,51,52,53]. Rad51 filaments replace RPA on ssDNA and spread into the surrounding dsDNA regions, thereby both eliminating the RPA-mediated stimulation and creating a steric hindrance for PCNA loading, which, in turn, impedes DNA synthesis [46,47,54]. Indeed, it has been shown that the Srs2-mediated disassembly of Rad51 nucleoprotein filaments is required for the efficient DNA synthesis during DSB repair by a number of different mechanisms [46]. Both, the initiation and net re-synthesis rates of resected DSBs are significantly reduced by *SRS2* deletion [46,55]. However, in *srs2Δ* cells, Rad51 nucleoprotein filaments are still affected by Rad54 which can assist in strand invasion and thus stimulate the resolution of Rad51 nucleoprotein filaments by helping them to progress to the downstream steps of HR [1,9]. In addition, Rad54 has a documented ability to remove Rad51 from dsDNA in vitro [56] and is required for efficient resolution of damage-associated Rad51 foci in vivo [57]. It appears that Rad54 activities in first stimulating Rad51 foci formation and then promoting their resolution are controlled by phosphorylation of Rad54 which occurs in late G2 and increases the ability to strip Rad51 from dsDNA [57,58]. Rad54 can also contribute to counteracting the unspecific binding of Rad51 to undamaged chromatin, but in yeast, this role is mostly carried out by another SWI/SNF translocase Rdh54 [59,60]. 

*S. cerevisiae* mutants lacking both Srs2 and Rad54 are not viable unless Rad51 or its mediators are removed [61,62]. This suggests that the assembled Rad51 nucleoprotein filaments without their regulation by Srs2 and Rad54 might be toxic. In this work, we overcame the synthetic lethality of *srs2Δ* and *rad54Δ* by creating conditionally viable strains, which then allowed us to analyse *srs2Δ rad54Δ* cells and address the role of the regulation of Rad51 nucleoprotein filaments in DNA replication and repair, and its importance for genome stability and cell survival.

## 2. Materials and Methods

### 2.1. Yeast Strains, Oligonucleotides and Plasmids

Yeast strains and oligonucleotides used in this study are described in Appendix A, respectively.

### 2.2. Spotting Assay

Strains with the *P_GAL1_-RAD55* construct were pre-grown on YPRAF plates at 30 °C overnight to derepress the *P_GAL1_* promoter. The next day, cells were resuspended in YPRAF broth at OD_600_ = 5.5-fold sequential dilutions of the suspensions that were made in a 96-well plate, and a 48-pin frogger was then used to transfer approximately 4 µL of cell suspension from each well onto YPGAL and YPD plates. Cells were incubated at 30 °C until appropriately sized colonies were formed for the most diluted samples. The images of the colonies were taken using Gel Doc XR+ imaging system (BioRad, Watford, UK).

### 2.3. SSA Plating Assay 

Strains with the SSA system were pre-grown on YPRAF plates at 30 °C overnight. The next day, cells were resuspended in YPRAF broth and the appropriate serial dilutions were plated on YPD and YPGAL plates. The plates were incubated at 30 °C until appropriately sized colonies were formed. To distinguish between the cells that have repaired the break by SSA and NHEJ, the colonies formed on YPGAL plates were replica-plated on YPD + G418 plates. The frequency of DSB repair by SSA was then calculated as the ratio of G418-sensitive colonies from the YPGAL plates to the total number of colonies on the corresponding YPD plates.

### 2.4. Treatment with α-Factor and P_GAL1_ Induction during Synchronised Growth Experiments

Cells were pre-grown on YPRAF plates at 30 °C overnight and then used to start 10 mL cultures in YPRAF broth. The broth was supplemented with 100 μg/mL of G418 for the strains with the SSA system to eliminate the cells that might have prematurely induced DSBs and repair them by SSA. The cultures were grown at 30 °C with aeration for approximately 8 h while being maintained in the log phase. The cultures were then diluted in YPRAF broth (with 100 μg/mL of G418 for the strains with the SSA system) so that the yeasts would reach the desired OD_600_ the next morning and incubated at 30 °C with the appropriate aeration. When the optical density of around 0.3 at OD_600_ was reached, α-factor (peptide sequence: WHWLQLKPGQPMY, Peptide Protein Research Ltd., Bishops Waltham, UK) was added to the final concentration of 5 μg/mL (*BAR1* cells) or 0.01 μg/mL (*bar1Δ* cells) to synchronise the cells in G1. After 2.5 h, the cultures were supplemented with galactose (*P_GAL1_* induced) or raffinose (*P_GAL1_* not induced) to the final concentration of 2% (*w/v*). After an hour, cells were pelleted, washed with YP, and transferred into fresh media with the appropriate carbon source to release from the G1 arrest. Nocodazole was added to the final concentration of 15 μg/mL right away to stop the cell cycle at G2/M, or α-factor was added to the final concentration of 0.01 μg/mL (*bar1Δ* cells) 50 min after the initial release to stop the cell cycle at the next G1. The cultures were incubated at 30 °C with the appropriate aeration. 

### 2.5. Southern Blotting Analysis of the Repair Product Formation during SSA

Probes used for the Southern blotting experiments were labelled using a random prime labelling kit (Prime-It II, Agilent Technologies, Santa Clara, CA, USA, 300385) and α-^32^P-dATP (Perkin Elmer, BLU012H250UC). Signal quantification was performed using phosphor-storage screens, the Typhoon FLA 7000 IP2 imager (GE Healthcare, Chalfont St Giles, UK) and the ImageQuant TL software (GE Healthcare, version 8.1).

Genomic DNA extracted from the samples collected during the time-course experiments was digested with EcoRI (NEB, Ipswich, MA, USA, R0101L) and SalI (NEB, Ipswich, MA, USA, R0138L) restriction enzymes, resolved on 0.6% TBE agarose gels (3.7 V/cm for 7 h), transferred to a positively charged nylon transfer membrane (Amersham HybondTM-N+, GE Healthcare, Little Chalfont, UK, RPN303B), and analysed by Southern blotting. A probe that hybridises to a region of the *URA3* gene (amplified using OSM2161 and OSM2162) was used to detect the fragments originating from the locus with the inducible DSB and track the formation of the SSA repair product. A separate probe (amplified using OSM189 and OSM190) that hybridises to the *ARS1* locus was employed to detect the reference fragments which were used to normalise the DNA in different samples. The relative amounts of SSA repair product were then normalised to the −1 h time point (prior to the DSB induction, corresponding to 100% of cells) for each strain separately to calculate the fraction of the cells that formed the repair product.

### 2.6. qPCR Analysis of Non-Homologous End Cleavage during SSA

qPCR reactions were performed using Brilliant II SYBR^®^ Green QPCR Master mix (Agilent, Waldbronn, Germany, #600828) and X3000P Real-Time thermocycler (Agilent Technologies, Waldbronn, Germany). Every DNA sample acquired from the time-course experiments was run in triplicates for each pair of primers. The kinetics of non-homologous end cleavage during SSA was quantified with the primers OSM2233 and OSM2234 which amplify the sequence that spans one of the cleavage sites. A second set of the qPCR reactions performed with the primers OSM1006 and OSM1007 which are specific to the reference locus *ARO1* was used to normalise the DNA from different samples. The fraction of yet uncleaved non-homologous DNA ends was then normalised to the 0 h time point (prior to the release from α-factor arrest, corresponding to 100% of cells as cleavage cannot happen before DNA end resection which, in turn, does not start until cells enter S phase) for each strain separately. The resulting numbers were subtracted from 1 and presented as a fraction of the cells that have cleaved the non-homologous DNA ends during SSA.

### 2.7. Cell Fractionation

Strains were pre-grown and synchronised in G1 before *P_GAL1_* was induced as described above. The progression of the cell cycle was stopped at G2/M using 15 μg/mL of nocodazole. A total of 2 h after the release from the G1 block, cells were diluted to OD_600_ = 0.5 with fresh prewarmed media and formaldehyde was added to the final concentration of 1.4% to the cultures incubated at 30 °C in a shaking water bath to crosslink proteins to DNA. Crosslinking was stopped after 10 min by adding glycine to the final concentration of 0.25 M and incubating the cultures for 5 min at 30 °C with the appropriate shaking. Aliquots equal to 3 and 20 OD_600_ units of cells were then collected for each culture in order to perform trichloroacetic acid (TCA) protein precipitation and cell fractionation, respectively. 

For the TCA protein precipitation, cell pellets were resuspended in 150 µL of lysis solution (1.85 M NaOH and 7.4% (*v/v*) β-mercaptoethanol) and incubated on ice for 10 min. An equal volume of 50% TCA solution (4 °C) was then added, and the samples were incubated on ice for an additional 10 min. Precipitated proteins were pelleted by centrifugation at 20,817× *g* for 2 min at 4 °C, washed with cold (−20 °C) 100% acetone, and resuspended in 1× sample buffer (50 mM Tris-HCl, pH 6.8, 100 mM DTT, 2.5% (*w/v*) SDS, 10% (*v/v*) glycerol, and 0.025% (*w/v*) bromophenol blue). Samples were incubated at 99 °C for 30 min in a PCR machine to de-crosslink the proteins from DNA.

For the cell fractionation, cell pellets were resuspended in 260 µL of the bead-beating buffer (20 mM Tris-HCl, pH 7.4, 150 mM NaCl, 1 mM PMSF, 1 cOmplete Mini, EDTA-free Protease Inhibitor Cocktail tablet (Roche, 11836170001, Mannheim, Germany) per 5 mL of buffer), an equal volume of glass beads (0.5 mm diameter, BioSpec Products, Bartlesville, OK, USA) was added, and cell walls were broken by 3000 rpm vortexing (25 cycles: 30 s of vortexing, 1 min of resting on ice). Cells were then separated from the beads and lysed with 1% (*w/v*) N-Lauroylsarcosine for 10 min on ice. The debris and unbroken cells were then removed by centrifugation at 425× *g* for 2 min at 4 °C and the resulting supernatants were centrifuged at 20,817× *g* for 15 min at 4 °C to pellet the crosslinked chromatin. Chromatin was then washed with the wash buffer (20 mM Tris-HCl, pH 7.4, 150 mM NaCl, 1% (*w/v*) N-Lauroylsarcosine sodium salt, 1 mM PMSF, 1 cOmplete Mini, EDTA-free Protease Inhibitor Cocktail tablet (Roche, 11836170001, Mannheim, Germany) per 5 mL of buffer) and resuspended in 100 µL of the same buffer. The suspensions were sonicated in Bioruptor Plus sonicator (Diagenode, Ougrée, Belgium) for 10 cycles (30 s ON/30 s OFF) on the HIGH power setting to fully solubilise the crosslinked chromatin. Samples were incubated at 99 °C for 30 min in a PCR machine, and then mixed with 4× sample buffer (200 mM Tris-HCl, pH 6.8, 400 mM DTT, 10% (*w/v*) SDS, 40% (*v/v*) glycerol, 0.1% (*w/v*) bromophenol blue) to the final concentration of 1× and boiled before being loaded onto SDS-PAGE.

### 2.8. Western Blotting

Protein extracts for Rad51 and actin Western blotting were resolved in 11% SDS polyacrylamide gel and transferred onto PVDF transfer membranes (Immobilon^®^-FL, 0.45 μm pores, Merck Millipore Ltd., IPFL00005, Darmstadt, Germany), while those for the histone H2B Western blotting were run in 15% SDS polyacrylamide gel and transferred onto nitrocellulose blotting membranes (Protran BA83, 0.2 µm, Whatman, Florham Park, NJ, USA). Samples for the analysis of Rad53-13Myc hyperphosphorylation were resolved in 5% SDS polyacrylamide gel and transferred onto PVDF transfer membranes (Immobilon^®^-FL, 0.45 μm pores, Merck Millipore Ltd., IPFL00005, Darmstadt, Germany). Anti-Rad51 (y-180) rabbit polyclonal (Santa Cruz Biotechnology, Dallas, Texas, USA, sc-33626, 1:200 dilution), anti-β-actin mouse monoclonal (Abcam ab8224, Cambridge, UK, 1:50,000 dilution), anti-Histone H2B (yeast) rabbit monoclonal (Abcam, Cambridge, UK, ab188291, 1:2000 dilution), anti-c-Myc, and mouse monoclonal (Thermo Fisher Scientific, Loughborough, UK, (13-2500), 1:1000 dilution) antibodies were used for Rad51, actin, histone H2B, and Rad53-13Myc Western blotting, respectively. Goat anti-mouse IgG (H + L) cross-adsorbed the secondary antibody, Alexa Fluor 680 (Thermo Fisher Scientific, Loughborough, UK, A-21057, 1:12,500 dilution) and goat anti-rabbit IgG F(c) IRDye800 Conjugated (Rockland, Limerick, PA, USA, 611-132-003, 1:12,500 dilution) secondary antibody were used with the corresponding primary antibodies. Western blotting membranes were scanned using Odyssey^®^ CLx fluorescent scanner (LI-COR^®^, Cambridge, UK). The resulting images were analysed and the proteins in the samples were quantified using Image Studio™ Lite software (version 5.2).

### 2.9. Rad51 Chromatin Immunoprecipitation and DNA Sequencing (ChIP-seq)

Samples for the *P_GAL1_-RAD55 srs2Δ rad54Δ* double mutants and the *P_GAL1_-RAD55* controls were collected the same way as described for the cell fractionation, except 200 OD_600_ units of cells were harvested. Samples for the NK1325 (strain with an HO-inducible unrepairable DSB, used as a control for the Rad51 ChIP-seq protocol) were collected by pre-growing a culture in YPRAF to OD_600_ = 0.4, then split into two, and galactose (induced) was added to one of them for the final concentration of 2% (*w/v*), and the same volume of raffinose (uninduced) was added to the other one. Cultures were incubated at 30 °C with aeration for 2 h and then crosslinked as described for the cell fractionation protocol.

The 200 OD_600_ units harvested for each strain were split into four Eppendorf tubes (50 OD_600_ units in each) and resuspended in 400 µL of bead-beating buffer (50 mM HEPES-KOH pH 7.5, 150 mM NaCl, 1 mM EDTA, 1 mM PMSF, 1 cOmplete Mini, EDTA-free Protease Inhibitor Cocktail tablet (Roche, 11836170001, Mannheim, Germany) per 5 mL of buffer). An equal volume of glass beads (0.5 mm diameter, BioSpec Products, Bartlesville, OK, USA) was added, and the tubes were vortexed for 25 cycles (3000 rpm; 30 s of vortexing, 1 min of resting on ice). After vortexing, cells were separated from the beads and cells and nuclei were lysed with 0.5% (*w/v*) SDS for 10 min on ice. Tubes were then centrifuged at 20,817× *g* for 15 min at 4 °C, the resulting supernatants were removed, the pellets were washed with 1 mL of 0.1% SDS buffer (50 mM HEPES-KOH pH 7.5, 150 mM NaCl, 0.1% (*w/v*) SDS, 1% Triton X-100 (*w/v*), 0.1% Sodium Deoxycholate, 1 mM EDTA, 1 mM PMSF, 1 cOmplete Mini, EDTA-free Protease Inhibitor Cocktail tablet (Roche, 11836170001, Mannheim, Germany) per 5 mL of buffer), and resuspended in 300 µL of the same buffer. Cell extracts were then sonicated in Bioruptor Plus sonicator (Diagenode, Ougrée, Belgium) for 20 cycles (30 s ON/30 s OFF) on the HIGH power setting. After the sonication, cell debris was spun down at 20,817× *g* for 15 min at 4 °C, and supernatants were transferred into fresh Eppendorf tubes containing 1 mL of the 0.1% SDS buffer. Samples belonging to the same strain (split into 4 Eppendorf tubes at the beginning) were then pooled together. A total of 10 µL were collected as input aliquots for each sample, mixed with 390 µL of TE (10 mM Tris-HCl, pH 8, 1 mM EDTA), and stored at −80 °C. Four immunoprecipitation reactions were set up for each sample by mixing 1 mL of sonicated cell lysate with 15 µL of Anti-Rad51 (y-180) rabbit polyclonal (Santa Cruz Biotechnology, Dallas, Texas, USA, sc-33626) antibody and 15 µL of Dynabeads protein A magnetic beads (Invitrogen, Paisley, UK, 10002D). The mixtures were incubated overnight at 4 °C on a rotating wheel. The next day, beads were recovered using a magnet and sequentially washed with wash buffer 1 (50 mM HEPES-KOH pH 7.5, 275 mM NaCl, 0.1% (*w/v*) SDS, 1% Triton X-100 (*w/v*), 0.1% Sodium Deoxycholate, 1 mM EDTA, 1 mM PMSF, 1 cOmplete Mini, EDTA-free Protease Inhibitor Cocktail tablet (Roche, 11836170001, Mannheim, Germany) per 5 mL of buffer), wash buffer 2 (50 mM HEPES-KOH pH 7.5, 500 mM NaCl, 0.1% (*w/v*) SDS, 1% Triton X-100 (*w/v*), 0.1% Sodium Deoxycholate, 1 mM EDTA, 1 mM PMSF, 1 cOmplete Mini, EDTA-free Protease Inhibitor Cocktail tablet (Roche, 11836170001, Mannheim, Germany) per 5 mL of buffer), wash buffer 3 (10 mM Tris-HCl, pH 8, 0.25 M LiCl, 1mM EDTA, 0.5% IGEPAL, 0.5% Sodium Deoxycholate), and wash buffer 4 (10 mM Tris-HCl, pH 8, 1 mM EDTA) for 5 min each wash on a rotating wheel at room temperature. To elute precipitated chromatin, beads belonging to the same sample were pooled together in 200 µL of TES (50 mM Tris-HCl, pH 7.5, 10 mM EDTA, 1% SDS) and then incubated at 65 °C for 10 min with mixing. Tubes were then vortexed, incubated at 65 °C for additional 10 min and the eluted chromatin was separated from the magnetic beads by centrifugation at 15,871× *g* for 3 min. The beads were washed with 200 µL of TE on the rotating wheel for 15 min at room temperature. The tubes were then centrifuged at 15,871× *g* for 3 min, and the resulting supernatants were mixed with the corresponding aliquots of the eluted chromatin.

To remove proteins from the IPed DNA, Proteinase K was added to the immunoprecipitated chromatin samples to the final concentration of 1 mg/mL, and the tubes were incubated at 65 °C overnight. The next day, immunoprecipitated DNA was purified using QIAquick^®^ PCR Purification Kit (Qiagen, Germantown, MD, 28106) and eluted in 35 µL of water.

ChIP-seq libraries were prepared using DNA SMART^TM^ ChIP-Seq Kit (Takara Bio, Mountain View, CA, USA, 634865) according to the manufacturer’s instructions and sequenced paired end on the Illumina MiniSeq sequencing system. The bioinformatic analysis of the sequencing data was performed using the Galaxy platform [63]. ChIP-seq reads were demultiplexed using Barcode Splitter (Galaxy Version 1.0.1) [64]. The reads were filtered for quality and trimmed using the PRINSEQ (Galaxy Version 0.20.4+galaxy1) [65] and Trim Galore! (Galaxy Version 0.6.7+galaxy0) [66] tools. Paired reads were then interleaved using seqtk_mergepe (Galaxy Version 1.3.1) [67]. Processed paired-end reads were aligned to the *S. cerevisiae* reference genome (sacCer3) using the Bowtie2 tool (Galaxy Version 2.4.2+galaxy0) [68]. Immunoprecipitated samples were normalized to the corresponding inputs using bamCompare (Galaxy Version 3.3.2.0.0) [69]. The relative enrichments of the Rad51 signal in *P_GAL1_-RAD55 srs2Δ rad54Δ* vs. *P_GAL1_-RAD55* and induced vs. uninduced NK1325 cells were then compared using bigwigCompare (Galaxy Version 3.3.2.0.0) [69] tool. 

### 2.10. FACS Analysis

Samples were collected during the time-course experiments described above by fixing around 0.6 OD_600_ units of cells with 1 mL of 100% ethanol at 4 °C overnight. The next day, cells were washed with 50 mM sodium citrate and vortexed at maximum speed for 30 s in 1 mL of the same solution. The wash step was then repeated, cells were resuspended in 1 mL of 50 mM sodium citrate containing 0.5 mg/mL of RNase A, and incubated at 37 °C overnight on a nutator. The next day, cells were washed twice with 50 mM sodium citrate and resuspended in 500 µL of the same solution containing 1 µM of SYBR™ Green I nucleic acid stain (Thermo Fisher Scientific, S7563). Samples were then incubated at RT for 1 h on a nutator in the dark. After the incubation, cells were sonicated using Bioruptor Plus sonicator (Diagenode, Ougrée, Belgium) for 10 cycles (30 s ON/30 s OFF) on the LOW power setting. Samples were analysed using the Attune™ NxT Flow Cytometer (Thermo Fisher Scientific, Loughborough, UK) and the FlowJo software (version X 10.0.7r2).

### 2.11. Live Cell Imaging

Yeast was pre-grown in YPRAF and synchronised in G1 before *P_GAL1_* was induced as described above. Cells were washed from the α-factor and transferred onto YPGAL agar pads and imaged live using Nikon Eclipse Ti2 inverted microscope equipped with a 100× 1.49 NA CFI Plan Apochromat TIRF objective, Spectra X light source (Lumencor, Beaverton, OR, USA), and a Prime 95B camera (Teledyne Photometrics, Tucson, AZ, USA). The imaging was carried out in the OKOLab Environmental chamber equilibrated at 30 °C, with the images taken every 5 min for at least 7 h in the brightfield, and mCherry channels with 9 Z-stacks of 0.5 μm. Available filters for the mCherry fluorophore channel were 575/25 for the SpectraX Illumination, 578/21 for the excitation filter, and 641/75 for the emission filter. The mCherry channel was set to 75 ms exposure time and 5% light source intensity. The images were analysed using Fiji (ImageJ) software (version 1.53t). A mitotic bridge was defined as a link of the mCherry signal between the two separating nuclei that persisted for two or more consecutive frames, i.e., longer than 10 min.

### 2.12. Transient RAD55 Expression and HU Treatment 

Cells were pre-grown and synchronised in YPRAF as described above. After 2.5 h of treatment with α-factor, cells were washed with YP and transferred into flasks with YP broth containing either raffinose or galactose, with or without 200 mM HU. After a 1 h incubation at 30 °C in a shaking incubator, each culture was diluted using YPD broth to stop *RAD55* expression and the appropriate dilutions were plated on YPD agar. Cells were incubated at 30 °C until appropriately sized colonies were formed. The HU survival frequencies in either raffinose or galactose were calculated as the ratios of the number of colonies grown after the 200 mM HU treatment to the number of colonies formed in a control sample with no HU added, both incubated in the media with the same carbon source.

### 2.13. Statistical Analyses

All statistical analyses in this study were performed using Student’s two-sample one-tailed t-test. If the difference between the standard deviation (SD) values of two averages being compared was less than two-fold–the equal variance *t*-test was used; otherwise, the unequal variance t-test was employed. *p* values were presented as follows: ns (*p* > 0.05), * (*p* ≤ 0.05), ** (*p* ≤ 0.01), *** (*p* ≤ 0.001).

## 3. Results

### 3.1. Rad54 and Srs2 Function to Provide Efficient DNA Synthesis during Repair 

The involvement of the Srs2 helicase in the disassembly of the Rad51 filaments has been well studied both in vivo and in vitro [42,43,44,45,46,70]. Although the molecular role of the Srs2-dependent Rad51 removal from chromatin during DNA repair has recently become clear [46], *srs2Δ* yeasts are viable. We hypothesised the existence of another protein with a similar role, and Rad54 with its previously reported ability to remove Rad51 from dsDNA was the most likely candidate [56]. Consistent with this hypothesis, *srs2Δ* and *rad54Δ* are synthetically lethal in a *RAD51*-dependent manner [61]. To address the dynamics of Rad51 nucleoprotein filaments in *srs2Δ rad54Δ S. cerevisiae*, the propagation of normally inviable *srs2Δ rad54Δ* mutants was enabled by constructing the strains with conditional *srs2Δ rad54Δ* lethality. To this end, *RAD55* was placed under a galactose-inducible promoter. When these *P_GAL1_-RAD55* strains were grown in YPRAF (*P_GAL1_* is not repressed by raffinose and can be easily induced by galactose addition) or YPD (glucose represses *P_GAL1_*), *RAD55* was not expressed and, thus, Rad51 filaments could not form efficiently, thereby suppressing the *srs2Δ* and *rad54Δ* synthetic lethality (Figure 1A). The expression of *RAD55* could be induced by adding galactose to the raffinose-containing media. Because Rad55 forms an obligate heterodimer with Rad57, the overexpression of *RAD55* from *P_GAL1_* is not expected to alter the abundance of the functional complex considerably. Although *P_GAL1_-RAD55 srs2Δ rad54Δ* were inviable in the presence of galactose (Figure 1A) and, therefore, could not be used to perform any genetic experiments relying on colony formation, the cells exposed to galactose for the duration of several h could be used for the molecular analysis and live cell imaging helping to understand how potentially unregulated Rad51 filaments might lead to cell death.

As mentioned above, the Srs2-dependent removal of Rad51 is required for efficient DNA synthesis during DSB repair [46]. A similar function of Rad54 in facilitating DNA synesis during repair was hypothesised and investigated here using the previously described genetic system consisting of *ura3-52* and *URA3* alleles flanking a *KAN* marker gene (confers resistance to the drug G418) and a recognition site for the HO-nuclease expressed from the *P_GAL1_* galactose-inducible promoter (Figure 1B,C) [46]. The DSBs induced in these cells upon the addition of galactose to the growth media were predominantly repaired by Single-Strand Annealing (SSA) using the homologies provided by the two *URA3* alleles.

SSA relies on Rad52, but neither Rad51 nor Rad55-Rad57 is required [1,71]. This allows using SSA as a system where Rad51 can have only a negative effect on the repair. Indeed, Rad51 filaments can form on the resected DNA and interfere with SSA [46,72,73]. The efficiency of SSA in different genotypes was first measured using a plating assay. Consistent with the published findings, *P_GAL1_-RAD55 srs2Δ* cells had a significant defect in DSB repair by SSA (Figure 1D). The SSA efficiency values in the *P_GAL1_-RAD55 srs2Δ* mutants observed in this setting were quite similar to those previously determined in *RAD55 srs2Δ* cells using the same genetic system (65 ± 4% vs. 58 ± 11%, respectively) [46]. This suggests that the *RAD55* expression from the exogenous promoter does not alter the balance of the HR factors significantly, likely because the level of the Rad55-Rad57 complex which functions as an obligate heterodimer remains largely unchanged when *RAD55* is overexpressed [24,26,27,74].

In contrast to *P_GAL1_-RAD55 srs2Δ*, *P_GAL1_-RAD55 rad54Δ* repaired as efficiently as the *P_GAL1_-RAD55* control strains (Figure 1D). The *P_GAL1_-RAD55 srs2Δ rad54Δ* cells could not be examined by plating as they do not form colonies on galactose-containing plates. Nonetheless, the progress of the DSB repair by SSA could be monitored in these strains by DNA analysis using Southern blotting [46]. The strains of interest were pre-grown in raffinose-containing media to the early log phase and synchronised with α-factor in G1 for 2.5 h. Galactose was then added to the G1 cells to induce the expression of both *HO* and *RAD55*. After another hour, cells were washed from the α-factor and released into fresh galactose-containing media with nocodazole. This allowed them to progress to the S phase, where they started repairing the HO-induced DSBs, but arrested in G2 due to the nocodazole added in order to prevent further propagation of the cells with the breaks repaired. Cell aliquots were collected every hour beginning from the galactose addition and the DNA was analysed by Southern blotting to quantify the appearance of the repair products formed after the DSB repair by SSA (Figure 1B,E).

Similar to the previously published data [46], the *SRS2* deletion resulted in a significant decrease in the repair product formation in the *P_GAL1_-RAD55* background (Figure 1F). The loss of Rad54 might have also negatively affected the dynamics of SSA as indicated by the apparently slower product formation compared to the *P_GAL1_-RAD55* control. However, these differences have not passed the threshold of statistical significance and the efficiency of the repair product formation in *P_GAL1_-RAD55* and *P_GAL1_-RAD55 rad54Δ* strains were indistinguishable by the end of the experiment (Figure 1F, 4 h after the G1 release). In contrast, the *srs2Δ rad54Δ* double mutants showed a major defect in the repair. A small fraction of the cells generated the repair product within the first hour of the post-G1 release, perhaps when the resected ends were annealed to each other before the further extended resection and Rad51 accumulation could take place. However, no significant increase in the repair product was detected over the next 3 h. This suggests that Rad54 participates in the DSB repair by SSA along with Srs2.

SSA consists of multiple steps, a defect in any of which could result in a decreased repair product formation observed in the *P_GAL1_-RAD55 srs2Δ* and *P_GAL1_-RAD55 srs2Δ rad54Δ* strains (Figure 1F). For the repair product analysed by Southern blotting in Figure 1 to form, (I) the DNA around the DSB needs to be resected, (II) the homologous sequences are to be annealed, (III) the non-homologous DNA ends are to be cleaved off, and (IV) the DNA re-synthesised at least to the EcoRI and SalI restriction sites used in the Southern blotting experiments (Figure 1C). To identify the affected steps, the non-homologous end cleavage (step III) was monitored through the time course using qPCR with the primers which amplify the area spanning the cleavage site of the non-homologous DNA ends (Appendix A) [46]. However, there was no significant difference between the *P_GAL1_-RAD55* control and the *P_GAL1_-RAD55* cells lacking Srs2 and/or Rad54 (Appendix A), suggesting that up to this point, neither *srs2Δ* nor *rad54Δ* affected the progress of SSA. Therefore, the difference between the strains observed in Figure 1F was due to a downstream step, namely DNA synthesis. This conclusion resembles the one previously published for Srs2 [46] and suggests that Rad54 also functions at the latest stage of SSA, DNA synthesis. 

Srs2 facilitates SSA by removing Rad51 from resected DNA, and the repair defect in the *srs2Δ* cells is suppressed by *RAD51* deletion [46]. Similarly, the removal of Rad51 through deleting *RAD51* not only enabled the survival of *srs2Δ rad54Δ* cells but also restored the SSA efficiency to the wild-type levels (Appendix A) suggesting that Rad54 affects SSA through Rad51. Therefore, both Srs2 and Rad54 might be involved in the disassembly of the Rad51 nucleoprotein filaments required for efficient repair-associated DNA synthesis.

### 3.2. The Role of Rad54 in the Facilitation of DNA Re-Synthesis Can Be Revealed through Rad51 Overproduction

The *rad54Δ* single mutants do not have a clear defect in SSA repair (Figure 1D,F), perhaps because Srs2 on its own is sufficient to deal with the Rad51 nucleoprotein filament disassembly. In order to reveal the involvement of Rad54 in this process, cells were challenged with increased levels of Rad51. We overexpressed *RAD51* instead of *RAD55* (Appendix A) in order to boost the amount of DNA-bound Rad51, thereby increasing the need for the Srs2/Rad54-dependent regulation of Rad51. Indeed, the time-course experiments revealed a significant decrease in the repair product formation in the *P_GAL1_-RAD51 srs2Δ*, *P_GAL1_-RAD51 rad54Δ,* and *P_GAL1_-RAD51 srs2Δ rad54Δ* strains when compared to their *P_GAL1_-RAD55* equivalents, while the *P_GAL1_-RAD51* and *P_GAL1_-RAD55* controls showed no apparent difference (Figure 1F,G). 

As Rad51 has an inhibitory effect on the SSA pathway in general [72,73,75], the progression of the non-homologous end cleavage in the *P_GAL1_-RAD51* strains was assayed using the qPCR analysis as before [46]. The *P_GAL1_-RAD51* strains showed the same non-homologous end cleavage efficiency as the *P_GAL1_-RAD55* cells (Appendix A) suggesting that Rad51 overproduction *per se* did not affect the DSB repair in our experimental settings. In contrast, when the *RAD51* overexpression was combined with *srs2Δ* and/or *rad54Δ*, small but statistically significant decreases in the fractions of the cells with non-homologous DNA ends cleaved were observed by the end of the time-course experiment (Appendix A). This was likely due to fewer DSBs being channelled into the SSA pathway when the increased levels of Rad51 were combined with the impaired Rad51 removal machinery [72]. Nonetheless, the small decrease in the non-homologous DNA end cleavage detected in the *P_GAL1_-RAD51 rad54Δ* cells could account for only a very minor part of the much more substantial decrease in the efficiency of SSA observed in these mutants (Figure 1G). This indicates that the Rad51 overproduction mostly caused problems during the DNA synthesis step in SSA in the *rad54Δ* cells, providing further evidence that Rad54, such as Srs2, likely facilitates damage-associated DNA synthesis by promoting Rad51 removal from the DNA.

### 3.3. The C-Terminus of Srs2 Is Not Required for Its Role in Rad51 Nucleoprotein Filament Regulation in rad54Δ Cells

In S phase, Srs2 inhibits the formation of Rad51 nucleoprotein filaments at replication forks preventing deleterious unscheduled recombination events during replication. This function involves Srs2 recruitment to the forks via its interaction with SUMOylated PCNA [76]. However, the mutations in either PCNA or SUMO-interacting motifs (PIP and SIM, respectively) at the very C-terminus of Srs2 are not sufficient for the *srs2 rad54Δ* lethality [77,78]. We have also shown that Srs2 with a large C-terminal truncation up to the amino acid 860, Srs2(aa1-860), is sufficient for the role of Srs2 associated with the DNA synthesis during repair [46], implying that neither the putative Rad51-binding domain (aa875-902) nor the PIP-SIM is required for the Rad51 filament regulation. Here, we combined *srs2(aa1-860)* and *rad54Δ* in heterozygous diploid yeast, sporulated the cells, and dissected the tetrads (Appendix A). The *srs2(aa1-860) rad54Δ* spores were viable. Therefore, the role of Srs2 in the regulation of the Rad51 nucleoprotein filaments studied here can be genetically separated from its function in the suppression of recombination at the replication forks during the S phase as the latter relies on the Srs2 recruitment to the forks via the PIP–SIM interaction with PCNA–SUMO [76].

### 3.4. Chromatin-Bound Rad51 Accumulates in the Absence of Srs2 and Rad54

As demonstrated above, the Srs2- and Rad54-mediated regulation of Rad51 nucleoprotein filaments is required for DNA synthesis during DSB repair. It has been previously estimated that *S. cerevisiae* cells suffer only 0.12 spontaneous DSBs per cell cycle [79,80,81]. Although DSBs might not be frequent enough to be the sole reason for the *srs2Δ* and *rad54Δ* synthetic lethality, unregulated Rad51 nucleoprotein filaments might accumulate on chromatin containing other lesions, for example, paused and stalled replication forks, causing cell death in the absence of Srs2 and Rad54. A set of *P_GAL1_-RAD55* strains similar to the ones described above (Figure 1), but without the SSA system, was used to test if *srs2Δ rad54Δ* cells accumulated persistent Rad51 nucleoprotein filaments. To this end, the strains were pre-grown to the early log phase in raffinose-containing media, synchronised in G1 using α-factor, subjected to the *P_GAL1_* induction for one hour, and then released into fresh galactose-containing media with nocodazole. Two hours after the release from the α-factor arrest, proteins were crosslinked to DNA by adding formaldehyde to the cultures. Cell samples were collected to quantify the Rad51 abundance in the chromatin fraction in comparison to the total cell protein (Figure 2). Because Rad52 is required for Rad51 filament formation, *rad52Δ* cells were used as a control for estimating the non-specific presence of Rad51 in the chromatin fraction (Figure 2). While no statistically significant differences were observed in the total cellular Rad51 levels among all the strains, the *P_GAL1_-RAD55 srs2Δ rad54Δ* cells exhibited an increase in the relative amount of Rad51 present in their chromatin fraction after a single round of genome duplication (Figure 2B). At the same time, all the other strains had Rad51 in their chromatin at levels comparable to the one in the *rad52Δ* cells, i.e., at the level of non-specific DNA binding. This suggests that in the presence of either Rad54 or Srs2, very little or no Rad51 filaments accumulate on DNA. When neither Rad54 nor Srs2 is present, the level of Rad51 in the chromatin fraction increases and the accumulation of the nucleoprotein filaments becomes detectable. 

In order to test if Rad51 accumulated at any specific sites in the genome of *srs2Δ rad54Δ*, we performed a Rad51-specific ChIP-seq analysis on the *P_GAL1_-RAD55 srs2Δ rad54Δ* cells grown in a similar way as the cells used for the experiments in Figure 2. However, these experiments did not reveal any preferential accumulation of Rad51 throughout the genome. A control ChIP-seq sample derived from the cells with an unrepairable HO-induced DSB showed a clear peak of Rad51 accumulation at the break, as expected. Therefore, Rad51 might accumulate at random sites in the genome in the absence of Srs2 and Rad54. Alternatively, there might be a number of preferential loci for the Rad51 accumulation, but each of them has a rather low probability of Rad51 localisation in a given cell. Such Rad51 loci scattered throughout the population genomes would not produce enough protein enrichment over the background signal to be detectable by this approach.

### 3.5. Unregulated Rad51 Filaments Induce Cell Cycle Arrest, Mitotic Bridges and Aberrant Mitoses

To test the effect of the persistent Rad51 nucleoprotein filaments on the cell cycle progression, FACS analysis and Rad53-13Myc Western blotting were used to monitor the genome dynamics and DNA damage checkpoint activation in the *srs2Δ rad54Δ* synchronised populations as the cells advanced through the cell cycle. *P_GAL1_-RAD55 srs2Δ rad54Δ rad53-13Myc* cells were pre-grown in raffinose-containing media to the early log phase, arrested at G1 using α-factor, and split into two cultures (Figure 3A). Galactose was added to one of the cultures (designated as induced) to activate the expression of *RAD55* from the *P_GAL1_* promoter, thereby enabling the formation of Rad51 nucleoprotein filaments and initiating the processes leading to cell death in the *srs2Δ rad54Δ* background. An equal amount of raffinose was added to the other culture (designated as uninduced) which was used as a control. One hour later, cells were washed from the α-factor and transferred into fresh media with the appropriate sugar, either galactose or raffinose, allowing them to progress into the S phase. Samples for the FACS analysis and the Rad53 Western blotting were collected every 40 min for 6 h. Approximately 50 min after the release from the G1 arrest, fresh α-factor was added to arrest the cells at the beginning of the next cell cycle.

After the release from the α-factor arrest, both induced and uninduced *P_GAL1_-RAD55 srs2Δ rad54Δ* cells appeared to progress through the S phase without any noticeable difference and completed their bulk genome duplication in about 40 min, in line with the similar published experiments (Figure 3B) [82,83,84,85]. As expected, the majority of the uninduced control cells reached the next G1 within 2 h (Figure 3B, time point 2 h). In contrast, the induced *P_GAL1_-RAD55 srs2Δ rad54Δ* cells appeared to arrest at G2/M for at least 40 min (Figure 3B, time point 2$\frac{2}{3}$ h). Nonetheless, most of them also divided and reached the next G1 approximately 3 h and 20 min after the release from the α-factor arrest (Figure 3B, time point 3$\frac{1}{3}$ h). At this time point, a sub-G1 peak characteristic of the cells with either extensive DNA degradation or gross genome mis-segregations was detected in the FACS profiles of the induced culture, suggesting that the cells suffered further genome instability when transitioning from the G2/M to the following G1. Consistent with the FACS data, Rad53-13Myc hyperphosphorylation indicative of the activated DNA damage checkpoint was detected in the induced *P_GAL1_-RAD55 srs2Δ rad54Δ* cells after the bulk genome duplication and persisted until the end of the experiment. The Rad53 phosphorylation decreased at the later time points as more and more *srs2Δ rad54Δ* double mutants went through cell divisions (Figure 3C). Rad53 was not hyperphosphorylated in the uninduced control cells.

As indicated by the sub-G1 peak in the FACS analysis after the induction of the *RAD55* expression (Figure 3B), the *P_GAL1_-RAD55 srs2Δ rad54Δ* cells appeared to suffer significant amounts of genomic instability when transitioning from the first cell cycle to the following G1. This might be caused by abundant aberrant mitoses. To investigate this possibility, the nuclear dynamics in the *P_GAL1_-RAD55 srs2Δ rad54Δ* cells was followed using live-cell imaging. The histone H2B was tagged with the mCherry fluorescent protein, and the resulting strains were pre-grown in raffinose-containing media to the early log phase, synchronised at G1 using α-factor, and subjected to the *P_GAL1_* induction for one hour. Cells were then released from the α-factor arrest, placed on YPGAL agar pads, and imaged at 30 °C to observe the first cell division. As expected from the FACS experiments, the *P_GAL1_-RAD55 srs2Δ rad54Δ* cells arrested in G2. Interestingly, the post-arrest mitoses in the vast majority of the *srs2Δ rad54Δ* mutant cells were accompanied by mitotic bridges which were rarely observed in the control *SRS2 RAD54* cells (Figure 3D,E).

### 3.6. Unregulated Rad51 Nucleoprotein Filaments Hinder Cell Recovery from Replication Stress 

Mitotic bridges can be caused by unresolved HR intermediates prompted by a number of DNA lesions [86]. However, this was unlikely to be the case in the *P_GAL1_-RAD55 srs2Δ rad54Δ* mutants as Rad54 would be required to generate such HR intermediates [9,87]. Mitotic bridges could also be formed as a result of DNA under-replication or aberrant termination of DNA replication, both of which are symptomatic of the problems at replication forks [86,88,89,90]. Thus, the ability of the cells to recover from replication stress was investigated in the mutants lacking Srs2 and/or Rad54, in the absence and presence of Rad51 nucleoprotein filaments (Figure 4A). *P_GAL1_-RAD55* cells with *srs2Δ* and/or *rad54Δ* were pre-grown in the presence of raffinose to the early log phase, synchronised in G1 using α-factor, and then released into either raffinose- or galactose-containing media with or without 200 mM hydroxyurea (HU). After a one-hour incubation, cells from each culture were diluted and plated on glucose-containing media, thereby shutting off the *RAD55* expression and effectively diluting the HU away. The scored colonies were then used to calculate the survival of the HU treatment in YPRAF and YPGAL, and by comparing those to evaluate the effect of the Rad51 filament formation (through the *RAD55* expression in YPGAL) on the survival of the HU-generated replication stress in a particular genetic background (Figure 4A,B). 

In the absence of the *RAD55* expression, the lack of Srs2, Rad54, or both did not affect the HU survival as both the single and the double mutants recovered from HU as well as the *SRS2 RAD54* cells did (Figure 4B, left bars in each coloured pair). However, the *RAD55* expression caused a mild HU sensitivity of the *srs2Δ* mutants and significantly reduced the survival of the *srs2Δ rad54Δ* double mutants (Figure 4B, compare right and left bars in each coloured pair). These results suggest that in the absence of the downstream regulation by Srs2 and Rad54, Rad51 nucleoprotein filaments hinder the cell recovery from replication stress. 

## 4. Discussion

In this work, the *srs2Δ rad54Δ* synthetic lethality observed in *S. cerevisiae* cells was exploited to address the regulation of the Rad51 nucleoprotein filaments once they have been formed. In the absence of Srs2 and Rad54, unregulated Rad51 nucleoprotein filaments can strongly inhibit DNA synthesis during ssDNA gap filling as it has been shown using SSA as a model. DNA synthesis is a common late step shared by multiple repair pathways, which is necessitated by the generation of ssDNA regions by the resection nucleases. Therefore, the disassembly of the Rad51 nucleoprotein filaments by Srs2 and Rad54 might be required for efficient damage-associated DNA synthesis in general. This would be in line with the previously shown role of the Srs2-dependent regulation of Rad51 disassembly during SSA, break-induced replication, and de novo telomere addition [46]. 

The role of Rad54 in stimulating DNA synthesis after the formation of joint DNA molecules has been previously described in vitro [91]. Although the requirement of Rad54 for the DNA synthesis following strand invasion was also demonstrated in vivo [26], it was impossible to separate Rad54 functions in supporting the formation of mature recombination intermediates that can be used to initiate DNA synthesis, and the promotion of DNA synthesis itself. In our study, the repair of DSBs induced in cells with the SSA system did not require strand invasion, allowing us to investigate the role of Rad54 in the facilitation of damage-associated DNA synthesis specifically. The data presented in Figure 1 suggest that Rad54 can indeed facilitate DNA synthesis during DNA repair in vivo, but this function can be easily overlooked as it only becomes apparent in the absence of Srs2 or at higher than the wild-type levels of Rad51. 

Srs2 and Rad54 likely function in parallel as they can remove Rad51 from ssDNA and dsDNA, respectively [40,56]. Their combined activity can clear up dsDNA–ssDNA junctions at the sites of DNA damage, thereby not only eliminating the steric hindrance that Rad51 filaments might pose but also enabling the binding of RPA required for the efficient PCNA loading and the subsequent DNA synthesis (Figure 5) [1,46,47]. In *srs2Δ* cells, Rad54 could still remove Rad51 from dsDNA and bring the ends of the nucleoprotein filaments to the dsDNA–ssDNA junctions, potentially promoting the stochastic exchange between RPA and Rad51 monomers bound to the vicinal ssDNA [92]. This would explain why the *srs2Δ* mutants have only a partial defect in DNA repair. The vice versa could be expected in *rad54Δ* mutants, but the presence of Srs2 alone might be sufficient to provide efficient replacement of Rad51 with RPA and stimulate Rad51 dissociation from dsDNA at the junctions. In fact, both Srs2-mediated and stochastic disassembly of Rad51 nucleoprotein filaments might be enhanced in the absence of Rad54. As mentioned above, Rad54 promotes Rad51 binding to DNA during the early phases of the cell cycle [29,30,31,32,33,34]. Furthermore, Rad54 and its homologue Rdh54 have been shown to restrain Srs2 activity in vitro, hinting that they might restrict its ability to disassemble Rad51 nucleoprotein filaments to a limited time window in vivo, which might be important for the temporal regulation of HR [93]. Therefore, *RAD54* deletion likely makes Rad51 filament disassembly in *SRS2* cells more efficient. When *RAD51* is overexpressed, the equilibrium of Rad51 binding to DNA is pushed to the bound state increasing the stability of Rad51 nucleoprotein filaments. The higher load of Rad51 removal might then exceed the ability of Srs2 to efficiently promote damage-associated DNA synthesis, and the role of Rad54 in this process can be revealed in *rad54Δ* cells overexpressing *RAD51*, as shown in Figure 1G. In the *srs2Δ rad54Δ* double mutants, the dsDNA–ssDNA junctions remain largely covered by Rad51 nucleoprotein filaments which are evidently stable enough to block the loading of PCNA, thereby strongly inhibiting DNA synthesis (Figure 5).

Further examination of the processes affected by unregulated Rad51 nucleoprotein filaments has revealed that when their formation is permitted, the *srs2Δ rad54Δ* double mutants activate the DNA damage checkpoint, accumulate chromatin-associated Rad51, and undergo aberrant mitoses accompanied by a very high abundance of mitotic bridges. The checkpoint activation occurs much earlier than the cell divisions (Figure 3B,C), suggesting that initial replication problems might be responsible for this activation rather than the aberrant mitoses. The cells arrest in G2/M but in less than two hours (a single cell cycle duration in YPGAL), they escape the arrest and undergo cell divisions generating yeast with <1N DNA content. The inability to maintain a longer arrest might be a consequence of Rad51, replacing the vast majority of RPA at the problematic replication sites, thereby blocking the RPA-dependent recruitment of the DNA damage sensors, Mec1/Lcd1 and the 9-1-1 complex [94,95,96]. In agreement with this hypothesis, the activation of human CHK1, one of the major DNA damage checkpoint kinases phosphorylated by ATR (mammalian homologue of Mec1) is dampened by RAD51 overproduction [97]. Upon division, the *srs2Δ rad54Δ* cells appear to suffer further DNA damage which can be explained by the drastic increase in the formation of mitotic bridges observed in *srs2Δ rad54Δ*. Similar mitotic bridges have been previously detected in *rad54Δ* cells treated with MMS [98]. Mitotic bridges normally lead to DNA breaks and genome missegregations. The loss of a considerable part of the genome including at least some essential genes due to the missegregations as well as the further loss of the genetic material due to extensive degradation of broken DNA ends generated during aberrant mitoses are the likely causes of cell death in the *srs2Δ rad54Δ* double mutants.

But what is the origin of the mitotic bridges in the *srs2Δ rad54Δ* cells? HR is unlikely to happen without Rad54 which is required for the formation of stable joint molecule recombination structures leaving unresolved replication intermediates as the most likely cause behind the abnormalities observed during the cell division [9,26,35,36,37,38,39,87]. This suggests that unregulated Rad51 nucleoprotein filaments might cause problems during replication. Indeed, the absence of Srs2 and Rad54 significantly decreased the efficiency of cell recovery from a short transient HU-induced replication stress which had no detectable effect on wild-type cells.

Unregulated Rad51 nucleoprotein filaments might affect the recovery from replication stress by obstructing the restart of stalled replication forks. As discussed above, the *srs2* mutations disrupting the Srs2 interaction with PCNA-SUMO are not synthetically lethal with *rad54Δ,* raising a possibility that *srs2Δ rad54Δ* double mutants do not encounter significant problems at replication forks. However, the PCNA-mediated recruitment of Srs2 is generally counteracted at stalled replication forks in wild-type yeast cells by PCNA unloading [99], which likely enables the lesion bypass and replication fork restart by HR. This suggests that if Srs2 is needed at stalled replication forks after the formation of recombination intermediates, it would likely be recruited via pathways other than the interaction with PCNA, even in wild-type conditions. Srs2 is known to be recruited to DNA repair sites by NHEJ protein Nej1 [100]. In turn, Nej1 is recruited to damage sites by Ku which binds the ends of DNA DSBs [101]. It is not immediately obvious how Ku could bind stalled replication forks which typically do not have a dsDNA end. One possibility is the fork reversal. As mentioned earlier, fork reversal seems to be generally inhibited by the DNA damage checkpoint in *S. cerevisiae* as well as in its relative *Schizosaccharomyces pombe* [15,102]. However, it was previously demonstrated that in *S. pombe,* the resection of terminally arrested unbroken replication forks is regulated by Ku, implying that the processing of stalled replication forks involves a reversed fork intermediate with a dsDNA end, creating an opportunity for Srs2 to be recruited via the Ku-Nej1 pathway [103]. Overall, this argues that the lack of the synthetic lethality between *rad54Δ* and the *srs2* mutations disrupting the Srs2 interaction with PCNA-SUMO does not rule out the possibility that *srs2Δ rad54Δ* encounter problems at stalled replication forks and that wild-type cells Srs2 and Rad54 regulate Rad51 nucleoprotein filaments at the replication sites.

We propose two possible mechanisms explaining how Rad51 nucleoprotein filaments might impede the restart of stalled replication forks in the absence of Srs2- and Rad54-mediated regulation. The first model assumes that unregulated Rad51 nucleoprotein filaments impede the recovery from replication stress through the same molecular mechanism as they inhibit the damage-associated DNA synthesis, i.e., by blocking PCNA loading at the ss–dsDNA junctions. (Figure 6, model A). Although these stalled forks can be resolved by merging with the incoming active ones, stalling of two converging forks blocked by unregulated Rad51 nucleoprotein filaments would lead to an under-replicated region between these forks which would then hold the sister chromatids together during mitosis causing mitotic bridges [86,90].

The second model is centred around the role of Rad51 in fork reversal (Figure 6, model B). Fork reversal involves the unwinding of a normal replication fork with a three-way junction, and the formation of an X-shaped structure with a four-way junction where the newly synthesised strands are paired together [15]. As mentioned earlier, fork reversal seems to be largely inhibited in *S. cerevisiae* and, thus, it is mostly studied in human cells [15]. Despite not being able to catalyse fork reversal in vitro by itself, human RAD51 is required for this process in vivo [17,104]. The mechanism of RAD51 action during the fork reversal is still rather enigmatic as some evidence suggests that neither its strand-exchange activity nor stable nucleoprotein filament formation is essential [104]. RAD51 might facilitate the fork reversal by interacting with fork remodelling enzymes and/or capturing and stabilising the unwound nascent DNA strands [104]. Indeed, human RAD51 has been shown to stimulate the fork reversal activity of RAD54 in vitro [105]. Importantly, RAD54 can both regress and restore replication forks but RAD51-mediated stimulation seems to favour fork reversal [105]. 

By extrapolating the data from human cells, we suggest that fork reversal is a dynamic process largely skewed towards unreversed forks in normal yeast cells. In *srs2Δ rad54Δ* mutants, the fork reversal might still occur as two other *S. cerevisiae* fork remodelling enzymes Mph1 and Rad5 have been implicated in this task [98,106,107,108]. However, the fork restoration could be problematic in the absence of Srs2 and Rad54 as unregulated Rad51 nucleoprotein filaments might block the access of the remodelling enzymes to the regressed arms of the reversed replication forks, just like Rad51 nucleoprotein filaments block the access of the resecting nucleases to the DNA ends at the stalled forks [8,14]. Consistent with this, in vitro studies suggest that the binding of Rad5 to the end of a 3′ terminated strand at a replication fork is critical for its ability to restore the reversed fork to the original three-way junction, while this interaction is dispensable for the initial conversion to the reversed four-way structure [107]. Thus, it is possible that without the regulation by Srs2 and Rad54, persistent Rad51 nucleoprotein filaments at stalled replication forks shift the equilibrium towards the fork reversal by stabilising this state. Reversed forks are unlikely to properly merge with incoming forks, possibly forming complex structures with five-way junctions (Figure 6, model B). The replication termination problems would keep the sister chromatids physically connected and this could explain the mitotic bridges observed in *srs2Δ rad54Δ* mutants. 

Compared to our first model, the second one would require a lower frequency of replication fork stalling to cause aberrant mitoses as the junctions between the sister chromatids could be prompted by a single replication fork stalled at naturally occurring replication barriers such as those found in rDNA, tRNA genes, and telomeres [109]. However, in both models, the ensuing unresolved physical links between sister chromatids could lead to the formation of mitotic bridges and chromosome breakage.

Mammalian cells have two homologues of Rad54, RAD54L, and RAD54B [110,111], and five proteins known to be capable of removing RAD51 from ssDNA in vitro—BLM, FBH1, RECQL5, FANCJ, and PARI [112,113,114,115,116]. This suggests that the significance and implications of the regulation of the assembled Rad51 nucleoprotein filaments described in *S. cerevisiae* in this study are just as relevant in higher eukaryotes, if not more so. Consistent with this idea, it has been previously shown that the pharmaceutical stabilisation of RAD51 can cause lethality in some cancer cells expressing high levels of the recombinase [117]. This toxicity has been attributed to the formation of the RAD51 complexes on undamaged chromatin [117]. Our findings identified additional cellular processes that might be affected by cancer treatment strategies dependent on the stabilisation of RAD51 nucleoprotein filaments, namely the repair of DNA lesions, the restart of stalled replication forks, and the segregation of replicated DNA. Overall, this study demonstrates that the efficient disassembly of the recombination machinery is essential not only for DNA repair but also for cell proliferation and survival under normal growth conditions.

## Figures and Tables

**Figure 1 biomedicines-11-01450-f001:**
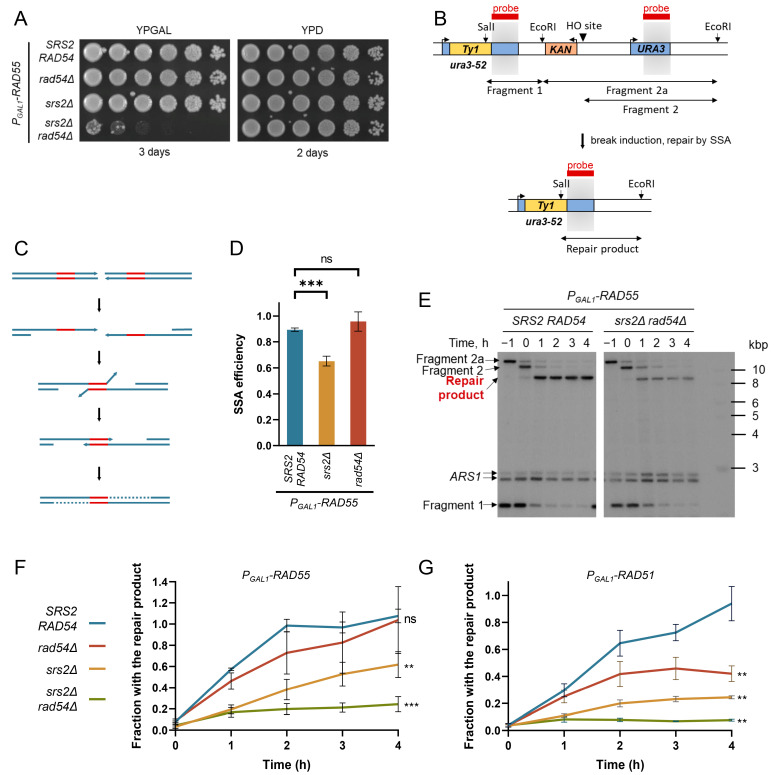
Rad54 and Srs2 are both required for DSB repair by SSA. (**A**) The conditional *srs2Δ* and *rad54Δ* synthetic lethality system. Cells pre-grown on YPRAF (YP + raffinose) were serially diluted (5× step) in YP and frogged on YPGAL (YP + galactose) and YPD (YP + glucose) agar plates. Strains used: NK6933, NK7200, NK7204, and NK7208. (**B**) SSA system before the induction of a DSB (top) and after its repair by SSA (bottom). Light grey boxes indicate the homologies predominantly used for SSA. As one of the direct repeats (grey boxes) used for the DSB repair and the sequence between the repeats are lost during SSA, the *KAN* marker in the SSA system allows to distinguish between SSA and Non-Homologous End Joining (NHEJ) repair events: *KAN* is always lost during the former but rarely during the latter. Double-headed arrows span the DNA fragments formed by the EcoRI + SalI restriction digest as well as the HO cleavage in vivo and monitored by the Southern blotting shown in panel E, using a probe (red boxes) that matches the homologies highlighted in grey. (**C**) A schematic of the DNA dynamics during SSA. After the initial resection of the DNA around a break, the direct DNA repeats are annealed to each other and non-homologous DNA ends are cleaved off, leaving two ssDNA gaps which are then filled in by DNA polymerases. Direct repeats are shown in red. The DNA synthesis at the post-annealing step is represented by dotted lines. (**D**) The SSA efficiency in *P_GAL1_-RAD55, P_GAL1_-RAD55 srs2Δ,* and *P_GAL1_-RAD55 rad54Δ* mutants determined by plating. Average ± SD of at least three biological repeats is shown for each genotype. Strains used: NK6724-NK6726; NK7291-NK7293; NK7424-NK7428; NK7188-NK7190. (**E**) Representative images of the Southern blots obtained using the samples collected during the time-course experiments analysed in panels (**F**,**G**). Blots were hybridised to two probes. One was specific to the *URA3* locus (see panel (**B**), red boxes) and allowed to monitor the DNA dynamics at the repair site on CHRV, while the other one hybridised to the *ARS1* locus on CHRIV and was used for normalisation. (**F**) Quantitative analysis of the SSA repair product formation over the time course in the *P_GAL1_-RAD55* background using Southern blotting. The average ± SD of at least three biological repeats is shown for each time point of each genotype. Asterisks describe the statistical significance of the difference between the value of the *P_GAL1_-RAD55* control and the value of each mutant derivative at the time point 4 h. Strains used: NK6724-NK6726; NK7188-NK7190; NK7291-NK7293; NK7295-NK7297. (**G**) Quantitative analysis of the SSA repair product formation over the time course in the *P_GAL1_-RAD51* background using Southern blotting. Same colour-coding was used as in panel (**F**). The average ± SD of at least three biological repeats is shown for each time point of each genotype. Asterisks describe the statistical significance of the difference between the value of the *P_GAL1_-RAD51* control and the value of each mutant derivative at the time point 4 h. The *p* values of the two-sample Student’s t-test are presented as follows: ns (*p* > 0.05), ** (*p* ≤ 0.01), *** (*p* ≤ 0.001). Strains used: NK5858-NK5860; NK5861-NK5863; NK5864-NK5866; NK5868-NK5870.

**Figure 2 biomedicines-11-01450-f002:**
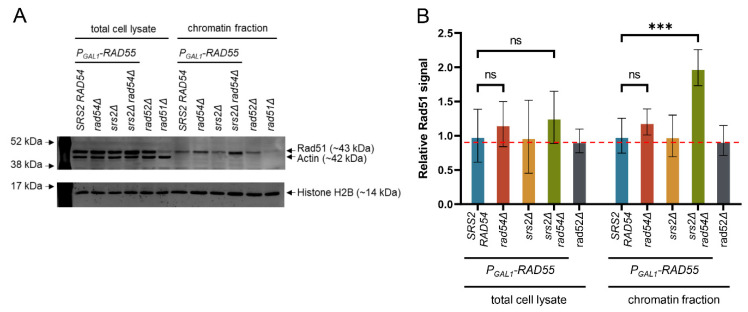
In the absence of Rad54 and Srs2, Rad51 accumulates on DNA. (**A**) Analysis of the Rad51 presence in the chromatin fraction by Western blotting. Actin was used as a non-chromatin protein marker and histone H2B as a chromatin marker. The faint band observed in the *rad51Δ* samples at the position where Rad51 normally runs comes from a cross-reacting protein (i.e., background) with a slightly lower gel mobility than Rad51. A representative set of Western blot images from one of the repeats. (**B**) A data summary plot for the relative Rad51 levels in the total cell lysates and chromatin fractions of the galactose-induced *P_GAL1_-RAD55 srs2Δ rad54Δ* cells and the appropriate control strains. Histone H2B was used as a normaliser to determine the relative amounts of Rad51 in different samples. The values were then normalised to the average value of *P_GAL1_-RAD55* strains for the total protein and the chromatin fractions separately. The average ±SD of three biological repeats is shown for each strain with the *P_GAL1_-RAD55* background and three technical repeats for the *rad52Δ* control. The red dotted line corresponds to the non-specific presence of Rad51 in the chromatin fraction based on the fact that Rad51 filaments cannot be formed in *rad52Δ* cells. The *p* values of the two-sample Student’s t-test are presented as follows: ns (*p* > 0.05), *** (*p* ≤ 0.001). Strains used: NK6933-NK6935; NK7200-NK7202; NK7204-NK7206; NK7208-NK7210; NK81.

**Figure 3 biomedicines-11-01450-f003:**
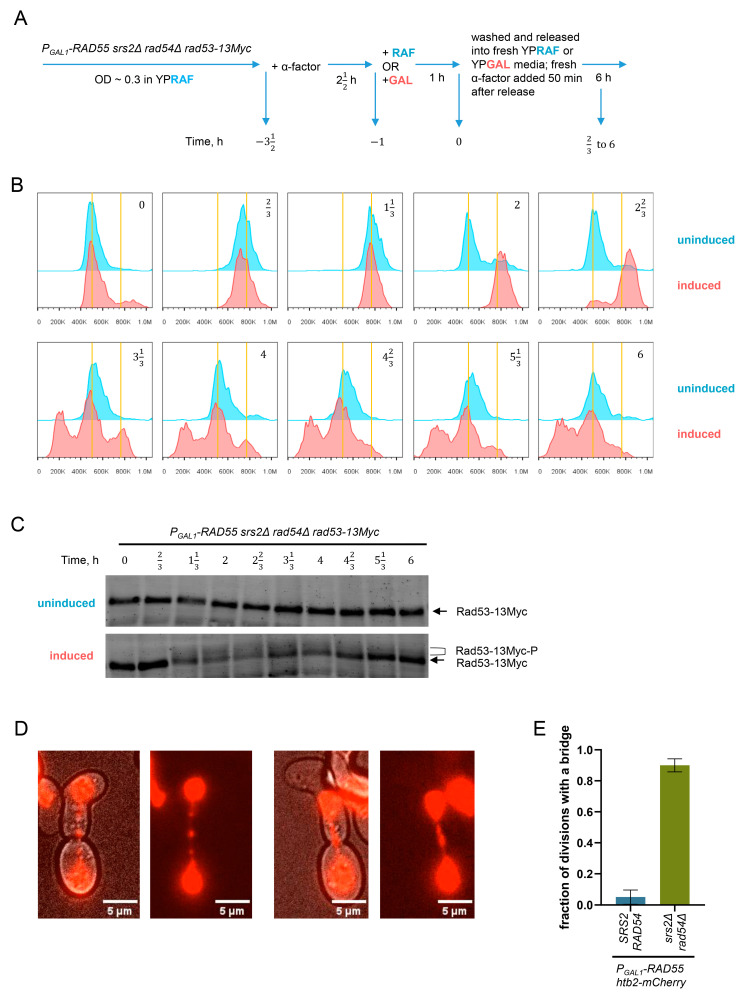
A single round of DNA replication in the cells with unregulated Rad51 results in the activation of the DNA damage checkpoint and aberrant mitoses. (**A**) A schematic of the time-course experiments performed to investigate the cell cycle progression and the DNA damage checkpoint dynamics in the *P_GAL1_-RAD55 srs2Δ rad54Δ rad53-13Myc* mutants. The time points correspond to the time (in h) before (negative) and after (positive) the release from the first α-factor arrest. (**B**) FACS profiles of the samples collected during the time-course experiment described in panel A. Vertical yellow lines specify the positions of the 1N and 2N peaks from left to right, respectively. Strains used: NK9134, NK9135. (**C**) Rad53 Western blotting analysis. The hyper-phosphorylation of Rad53 characteristic of the DNA damage checkpoint activation is seen in the galactose-induced *P_GAL1_-RAD55 srs2Δ rad54Δ* cells but not in the control culture grown in raffinose. The time points correspond to the ones in the panels (**A**,**B**). (**D**) Representative images of the mitotic bridges observed in the *P_GAL1_-RAD55 srs2Δ rad54Δ htb2-mCherry* cells grown in the presence of galactose. Two sets of images are shown. In each set, the image on the left shows the overlap of the bright field with the red channel while the image with the red channel on its own is on the right. (**E**) Quantification analysis of the nuclear divisions for the mitotic bridge formation. Only the first cell divisions after the *P_GAL1_-RAD55* induction were scored. The data bars show average ± SD based on three biological repeats for each genotype. A minimum of 75 mitoses were scored for each genotype. Strains used: NK10689-NK10691; NK10692-NK10694.

**Figure 4 biomedicines-11-01450-f004:**
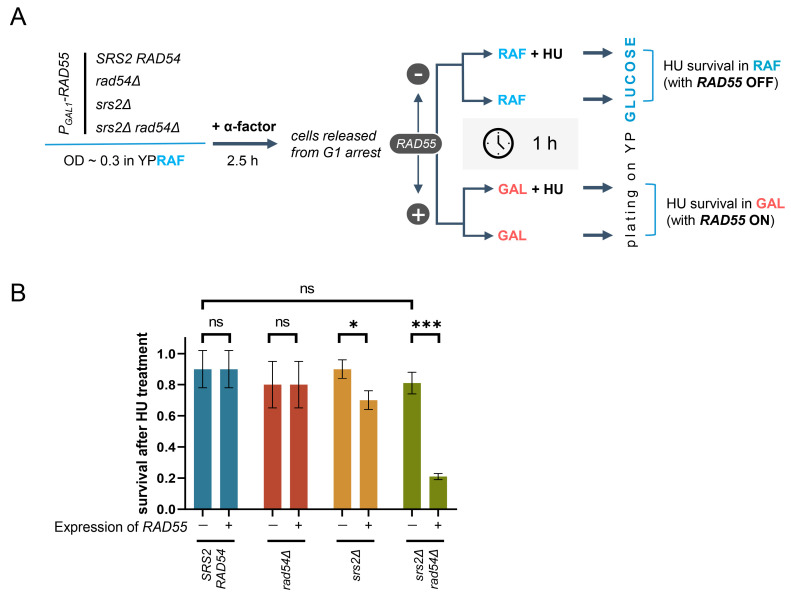
Presence of the Rad51 nucleoprotein filaments confers HU sensitivity in *srs2Δ* and *srs2Δ rad54Δ* yeast. (**A**) A schematic outline of the experiment testing cell recovery from the HU-induced replication stress, in the presence and absence of Rad51 nucleoprotein filaments. (**B**) Cell survival after a transient HU treatment with or without the expression of *RAD55.* The survival is calculated as the ratio of the colonies grown after the treatment either in raffinose (YPRAF + HU/YPRAF, left bars in each coloured pair) or in galactose (YPGAL + HU/YPGAL, right bars in each coloured pair). The data show average ± SD based on three biological repeats for each genotype. The p values of the two-sample Student’s t-test are presented as follows: ns (*p* > 0.05), * (*p* ≤ 0.05), *** (*p* ≤ 0.001). Strains used: NK6933-NK6935; NK7200-NK7202; NK7204-NK7206; NK7208-NK7210.

**Figure 5 biomedicines-11-01450-f005:**
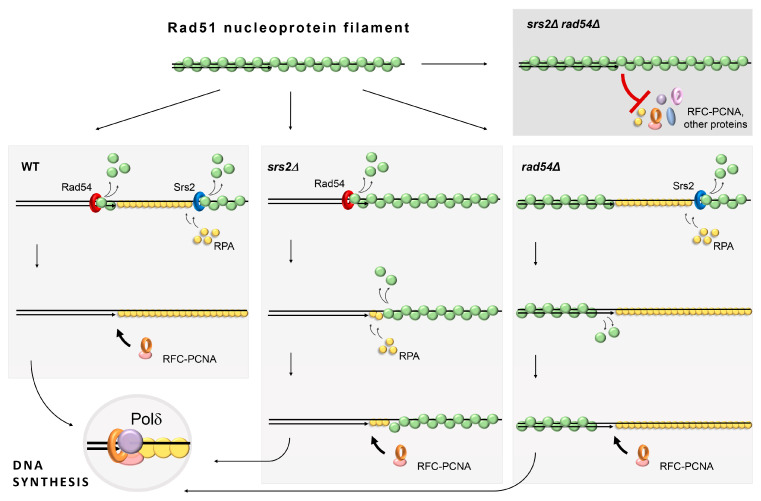
A model explaining the complementary functions of Srs2 and Rad54 in the damage-associated DNA synthesis.

**Figure 6 biomedicines-11-01450-f006:**
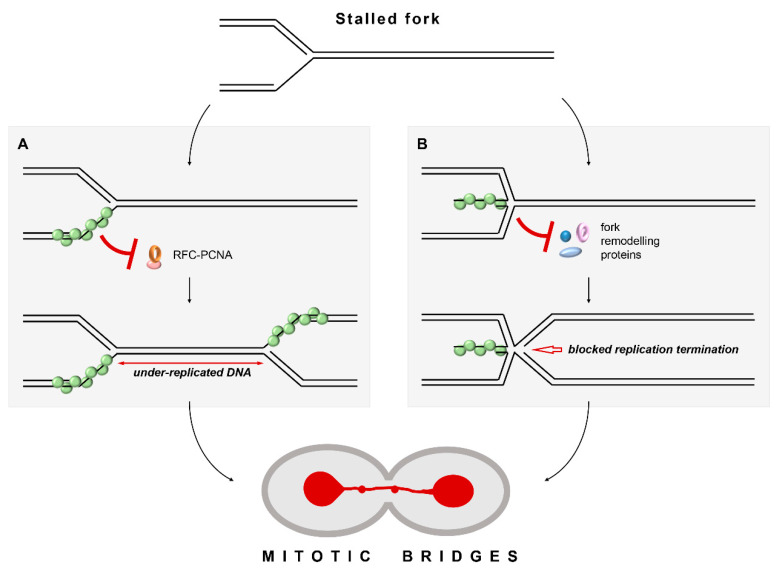
Two models (**A**,**B**) explaining how unregulated Rad51 nucleoprotein filaments might permanently stall challenged replication forks and cause mitotic bridges during cell divisions.

## Data Availability

The data from the Rad51-specific ChIP-seq experiment are openly available in the Gene Expression Omnibus (GEO) genomic data repository (https://www.ncbi.nlm.nih.gov/geo/) under the accession number GSE227383 (accessed on 16 March 2023).

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
