# Peer review of "The Inability to Disassemble Rad51 Nucleoprotein Filaments Leads to Aberrant Mitosis and Cell Death"

_biomedicines, 2023, doi:10.3390/biomedicines11051450_

Round 1
Reviewer 1 Report
In this study, the authors investigate the synthetic lethal interaction between srs2 and rad54. In vitro studies have shown that Srs2 and Rad54 remove Rad51 from single- and double-stranded DNA, respectively, suggesting that unresolved Rad51 nucleoprotein filaments are responsible for the lethal phenotype. Consistent with this idea, deletion of RAD51 or RAD55, which encodes a mediator of Rad51 nucleoprotein filaments, suppresses the synthetic lethal phenotype. The authors develop a system for conditional lethality of the srs2 rad54 double mutant by inducible expression of RAD55. Using a physical assay for single-strand annealing, they show a synergistic defect in formation of repaired products subsequent to flap removal, suggesting that fill-in DNA synthesis is prevented by unresolved Rad51 filaments. Even in unperturbed cells (no external DNA damage), Rad51 accumulates on chromatin in the absence of Srs2 and Rad54. The key experiment in the manuscript is shown in Figure 3, where they show that during an unperturbed synchronized cell cycle, srs2 rad54 cells arrest at G2/M and the DNA damage checkpoint in induced. Remarkably, most cells that transit through mitosis exhibit chromatin bridges, indicative of incomplete replication, and many daughter cells have a sub-1N DNA content.
The data presented are convincing and the manuscript is very well written. The only minor comment is that Rad51 nucleoprotein filament is generally used instead of nucleofilament.
Author Response
In this study, the authors investigate the synthetic lethal interaction between srs2 and rad54. In vitro studies have shown that Srs2 and Rad54 remove Rad51 from single- and double-stranded DNA, respectively, suggesting that unresolved Rad51 nucleoprotein filaments are responsible for the lethal phenotype. Consistent with this idea, deletion of RAD51 or RAD55, which encodes a mediator of Rad51 nucleoprotein filaments, suppresses the synthetic lethal phenotype. The authors develop a system for conditional lethality of the srs2 rad54 double mutant by inducible expression of RAD55. Using a physical assay for single-strand annealing, they show a synergistic defect in formation of repaired products subsequent to flap removal, suggesting that fill-in DNA synthesis is prevented by unresolved Rad51 filaments. Even in unperturbed cells (no external DNA damage), Rad51 accumulates on chromatin in the absence of Srs2 and Rad54. The key experiment in the manuscript is shown in Figure 3, where they show that during an unperturbed synchronized cell cycle, srs2 rad54 cells arrest at G2/M and the DNA damage checkpoint in induced. Remarkably, most cells that transit through mitosis exhibit chromatin bridges, indicative of incomplete replication, and many daughter cells have a sub-1N DNA content.
The data presented are convincing and the manuscript is very well written. The only minor comment is that Rad51 nucleoprotein filament is generally used instead of nucleofilament.
Thank you for pointing that out. We have changed all instances of “nucleofilament” to “nucleoprotein filament”.
Reviewer 2 Report
The authors examined how Srs2 and Rad54 regulate Rad51 binding to DNA and affect cell growth. Based on previous finding that synthetic lethality (SL) of srs2∆ rad54∆ is suppressed by rad51∆, a conditional SL system wherein Gal-Rad51 or Gal-Rad55 was used to induce acute lethality, allowing examination of molecular defects leading to SL. The authors focused on a SSA system which allows better examination of “toxic” effect of Rad51 bound DNA. The main findings include 1) srs2∆ rad54∆ increased chromatin-bound Rad51 (though ChIP-Rad51 did not reveal increased Rad51 peaks). 2) srs2∆ rad54∆ reduced SSA level and this reduction unlikely occurs at steps before DNA synthesis. 3) upon Rad51 overexpression, srs2∆ rad54∆ cells increased Rad53 levels and anaphase cells with DNA bridge, though S phase progressed normally. 4) upon Rad51 overexpression, srs2∆ rad54∆ leads to HU sensitivity. The authors conclude that inability of Rad51 dissociation from DNA due to loss of Srs2 and Rad54 lead to aberrant mitosis and cell death. Overall, the data presented are very clear and the paper was easy to follow. The findings support the overall conclusion. We have a few suggestions
1) Srs2 and Rad54 both are multi-functional proteins and alleles affecting specific features have been constructed in previous studies. Thus, it would be useful to examine whether some of these features are relevant to the srs2 rad54 SL. For example, srs2-PCNA binding mutant and Rad51 binding mutant can be used to ask whether these feature of Srs2 are important in preventing Rad54 SL.
2) Figure 3B. Rad53-13myc phosphorylation decreases in the induced sample at later time points. Please provide possible explanations for this observation.
3) Within the first hour, some repair products are detected in srs2Δ rad54Δ during SSA but nothing after. Can you provide an explanation for what was happening around the one-hour mark in these cells?
4) ChIP-seq data are said “not shown (Line 525)” but should be included to Figure 2 or supplemental figure.
Moderate editing will help to improve readability.
Author Response
The authors examined how Srs2 and Rad54 regulate Rad51 binding to DNA and affect cell growth. Based on previous finding that synthetic lethality (SL) of srs2∆ rad54∆ is suppressed by rad51∆, a conditional SL system wherein Gal-Rad51 or Gal-Rad55 was used to induce acute lethality, allowing examination of molecular defects leading to SL. The authors focused on a SSA system which allows better examination of “toxic” effect of Rad51 bound DNA. The main findings include 1) srs2∆ rad54∆ increased chromatin-bound Rad51 (though ChIP-Rad51 did not reveal increased Rad51 peaks). 2) srs2∆ rad54∆ reduced SSA level and this reduction unlikely occurs at steps before DNA synthesis. 3) upon Rad51 overexpression, srs2∆ rad54∆ cells increased Rad53 levels and anaphase cells with DNA bridge, though S phase progressed normally. 4) upon Rad51 overexpression, srs2∆ rad54∆ leads to HU sensitivity. The authors conclude that inability of Rad51 dissociation from DNA due to loss of Srs2 and Rad54 lead to aberrant mitosis and cell death. Overall, the data presented are very clear and the paper was easy to follow. The findings support the overall conclusion. We have a few suggestions
1) Srs2 and Rad54 both are multi-functional proteins and alleles affecting specific features have been constructed in previous studies. Thus, it would be useful to examine whether some of these features are relevant to the srs2 rad54 SL. For example, srs2-PCNA binding mutant and Rad51 binding mutant can be used to ask whether these feature of Srs2 are important in preventing Rad54 SL.
Thank you for bringing this up. In order to address this question, we have added an additional subsection in the Results section (lanes 506-622 if “Tracking Changes” option in Word is set to “No Markup”, lanes 512-528 if set to “All Markup”) and a paragraph in the Discussion (lanes 774-796 for “No Markup”, lanes 785-807 for “All Markup”)
2) Figure 3B. Rad53-13myc phosphorylation decreases in the induced sample at later time points. Please provide possible explanations for this observation.
We have added a sentence to highlight this in the Results section observation (lanes 632-634 for “No Markup”, lanes 640-642 for “All Markup”). A possible explanation for the observation can be found in the discussion (lanes 752-757 for “No Markup”, lanes 762-769 for “All Markup”)
3) Within the first hour, some repair products are detected in srs2Δ rad54Δ during SSA but nothing after. Can you provide an explanation for what was happening around the one-hour mark in these cells?
We have provided a possible explanation in lanes 448-450 for “No Markup”, lanes 454-456 for “All Markup”
4) ChIP-seq data are said “not shown (Line 525)” but should be included to Figure 2 or supplemental figure.
Thank you for the suggestion. Unfortunately, we were not able to visualise ChiP-seq data in a useful manner, as the results were negative. However, all sequences were deposited to an online repository. Specific details on how to access it can be found in the “Data Availability Statement” (lanes 878-880 for “No Markup”, lanes 896-898 for “All Markup”)
Comments on the Quality of English Language
Moderate editing will help to improve readability.
We introduced some changed to the text to improve readability, as suggested.